# An Adaptive Multi-Staged Forward Collision Warning System Using a Light Gradient Boosting Machine

**Jun Ma, Jiateng Li \*, Zaiyan Gong and Hongwei Huang**

School of Automotive Studies, Tongji University, No.4800, Cao-an Road, Shanghai 201804, China
\* Correspondence: 2011453@tongji.edu.cn

**Abstract:** The existing forward collision warning (FCW) systems that adopt kinematic or perceptual parameters have some drawbacks in the warning performance because of poor adaptability to the users or ineffectiveness of the warnings. To solve the problems of adaptability, several FCW models have been proposed based on algorithms (machine learning, deep learning). However, there is a lack of consideration for the multi-staged warning to avoid an abrupt warning that may startle or distract the driver. In this study, a light gradient boosting machine (LGBM) was adopted to develop a multi-staged FCW. The proposed model was trained and evaluated on a platform based on a driving simulator by twenty drivers. Through Shapley Additive Explanations (SHAPs), the output of the proposed model was explained. Specifically, the front vehicle acceleration, time-to-collision (TTC), and relative speed were found to strongly affect the warning stages from the proposed model. To evaluate the utility and acceptability of the developed model, it was compared with three existing FCW models in terms of subjective and objective indicators. As a result, a trade-off was found between the utility and user acceptance. Additionally, the comparison study also indicated that the developed model outperformed other previous models due to not only the high accuracy but also the suitable trigger timing for each participant.

**Keywords:** forward collision system; staged warning; light gradient boosting machine

## 1. Introduction

Recently, traffic accidents have caused huge casualties and economic losses, making them a serious problem for all countries. Vehicle-related accidents account for 86.8% of the number of traffic accidents in China in 2019 [1]. In addition, according to the National Highway Traffic Safety Administration (NHTSA), 36,560 people were killed in vehicle crashes, and forward collisions accounted for about 30% in 2018. Most of these collisions were caused by human factors [2]. The previous search has indicated that most crashes can be avoided if the drivers are alerted and are able to take steps within a second prior to the accidents [3]. Forward collision warning (FCW) systems were proven to be useful in helping drivers respond more quickly under emergencies [4,5]. Therefore, an effective FCW system plays a vital role in the improvement of road safety.

FCW systems are developed based on sensors or technology of V2V communication to provide drivers with warning messages so as to avoid rear-end collisions. Most of the FCW systems currently applied to mass production vehicles are based on sensor-based signals. These FCW algorithms are built based on fixed kinematic or [6] perceptual parameters. For instance, the Mazda FCW algorithm adopts the braking critical distance as the trigger parameter. However, the algorithm of fixed parameters is not suitable for every driver, because drivers have diverse perceptions about the coming dangers and driving experiences, and driving styles. As a result, an individualized FCW algorithm is desired. Machine learning, ensemble learning, deep learning, and other algorithms are widely used in FCW models [7–10]. However, the strategy of these proposed FCW models was designed as a single-level warning. Some previous studies indicated that a multi-staged warning

strategy is a more efficient way to remind drivers to prepare in advance without startling the drivers in the moment of emergency [11,12].

The aim of this paper was to develop an adaptive, staged FCW model based on a driving simulator. For driver adaptivity, a light gradient boosting machine (LGBM) was adopted to predict and provide the warning thresholds of staged FCW. The advantage of it is that the trigger timing of FCW matches the driver's perception and reaction to the coming safe-critical situation. Therefore, the proposed model is expected to be more acceptable for the drivers, since it is trained with the user's driving behavior data. For the warning stages, two levels of warning were designed based on the instruction of the handbook of human factors [13]. Then, the proposed model was evaluated for its utility using subjective and objective metrics, and its acceptability using a subjective questionnaire. Then, its results were compared with those of existing FCW models. The comparison not only aims at evaluating the proposed FCW model, but also at highlighting the importance of two designed guidelines (adaptivity and multiple stages of warning) to improve FCW models.

## 2. Literature Review

### 2.1. FCW Strategy Based on Fixed Parameters

The FCW strategy of most production vehicles is to use fixed perceptual parameters [6] or kinematic factors as the thresholds for a warning. For the perceptual approach, it provides the driver a warning in near-crash situations derived from the thresholds of perception. Time-to-collision (TTC) is widely adopted as a parameter to determine the trigger timing of warnings in many FCW models [14–16]. For instance, Wang established an FCW model based on TTC thresholds [17]. Moreover, the Honda's Collision Mitigation Braking System (CMBS) is a typical FCW and avoidance model based on the evaluation of the TTC [18]. However, previously established studies suggested that this algorithm may not be effective in all extreme cases [16,18]. As a result, late or missed warnings may distract or even startle the driver. For the kinematic approach, it provides warnings not only based on the fundamental parameters of the vehicle (vehicle velocity, acceleration, etc.), but also on the variables of human factors (brake reaction time, intentions, fatigue, etc.) [19]. Specifically, a threshold warning distance is defined from the functions of the aforementioned parameters. Additionally, previous studies utilized the warning distance as the warning threshold. The influences of braking parameters, road conditions, and driver factors are taken into account by altering the distance threshold while the algorithm is operating [20–22].

Mazda's algorithm provides a warning based on the braking critical distance [5]. Nevertheless, this algorithm is considered to be conservative because it attempts to avoid all collisions [23]. Therefore, it gives the driver the warning too frequently to make them desensitized to the subsequent warnings. Moreover, a previous study suggested that frequent warnings may exert a negative influence on a driver's performance due to excessive information [24]. In general, on one hand, there are differences in drivers' perceptual preferences in a car-following case: from cautious to aggressive. It leads to diverse timing of braking in the presence of the coming collision cases, hence the need for different FCW trigger timing [19,25]. On the other hand, a fixed warning threshold of critical distance is inappropriate for drivers with varied driving styles as well. As prior research has found differences in the perception of safe following distances for drivers with various driving styles [26]. As a result, the warning threshold of critical distance needs to be individualized. In summary, FCW models based on fixed thresholds of warning timing and distance are not adapted to the driving characteristics.

### 2.2. FCW Strategy Based on Algorithms

Recently, machine learning, ensemble learning, deep learning, and other algorithms have been widely used in FCW models [7–10]. Moreover, a combined SVM was used to establish a personalized autonomous lane change model [27]. A previous study adopted TTC as the output of warning prediction models [28]. The result indicates that LSTM

outperforms the DBN model and is suitable for car-following warnings. However, both the proposed models ignore the difference between individuals' perceptions of TTC. Therefore, the warning timing of the algorithms is not suitable for everyone. Pyo et al. adopted a CNN as the classifier and TTC as the trigger parameter to establish a vehicle-detection-based FCW system for highway environments [29].

Some models were trained using drivers' intentions or driving behavior data for improvements in warning accuracy. For instance, an ANN classifier was adopted to develop a collision warning model [9]. Specifically, the model is based on the front radar providing kinematic parameters (speed, acceleration, and relative distance) to a neural network classifier, and then it provides a warning to the driver approaching a collision. However, the above models have only focused on providing the driver with a single level of warning in emergency cases. In addition to the need to alert the driver in a safety-critical situation, previous studies suggest that the driver should also be given a prewarning of the danger of a possible collision in advance [11,12]. The multi-stage warning was proved to extend the warning range to more advanced warning levels of prewarning or elaborate the possible critical situation ahead [30]. Therefore, it ensures the driver stays focused on the road without providing a sudden warning that could startle or distract the driver.

To summarize, although the above models have contributed greatly to the diversity and development of FCW algorithms, they have either ignored the personalization of the driver's car-following preference or the importance of staged warnings. As a result, a preferable FCW design should be not only in line with the driver's driving preference but also designed to stage warnings to minimize false warnings and nuisance warnings. Therefore, an adaptive, multi-staged forward collision warning system is needed.

## 3. Materials and Methods

As shown in Figure 1, the general framework of this study consists of four parts: research aim, dataset construction, model development, and proposed model discussion. In this study, a driving characteristics acquisition test and FCW validation test were conducted. The aim of the driving characteristics acquisition test was to train and test the proposed FCW model. The driving operation data were collected to build a two-staged warning model according to the urgency of the impending collision. The objective of the FCW validation test was to evaluate the proposed FCW model. Simulation experiments were conducted for the above tests.

### 3.1. Driving Simulation System

The simulation system consists of an open cab and simulated scenarios. The open cab consists of Fanatec hardware (Fanatec, Landshut/Bavaria, Germany) and three monitors forming a 150° simulated view (Figure 2). The simulated scenarios were built via SCANeR studio® (AVSimulation, Boulogne-Billancourt, France). The total length of the two-way road is 8756.85 m. Twelve trigger points were randomly set along the route. The trigger points were randomly set along the straight roads. More specifically, the distance from each trigger point to the start of the specified road section was set randomly. If the trigger point is not set randomly, it is set to a fixed-distance spacing, for example. After repeated emergency braking events, participants might be prepared to brake in advance based on the trigger pattern due to the learning effect. As a result, the data we collect will not match the driver's habits. Finally, the trained FCW model is not adaptive to the driving patterns of the driver. Hence, random trigger points were adopted in the present study. When the front vehicle passes the trigger point, it immediately decelerates at 7 m/s2. That trigger design of forward collision warning was developed by script editing of SCANeR studio® (AVSimulation, Boulogne-Billancourt, France). Therefore, each driver will encounter trigger points at different locations on each lap. The advantage of this setup is to avoid a learning effect after the driver has experienced repeated exposure. The scenarios mainly include urban sections, suburban sections, and freeway sections; the driving scenarios mainly include a right turn at the intersection, a left turn at the intersection, and a straight ahead

at the intersection. The road sections with trigger points were also arranged in the above scenarios. The open-cab cockpit was modified based on a Fanatec® simulated cockpit (Fanatec ClubSport Wheel Base V2.5, ClubSport Pedal V3 Inverted, and RennSport Cockpit, Fanatec, Landshut/Bavaria, Germany). Three 43-inch screens were combined together to display the simulated driving scenarios, and they provided the drivers a 94° field of view. The open cab and the simulated scenarios were connected by the ACQUISITION module of SCANeR studio (AVSimulation, Boulogne-Billancourt, France). The signals from the steering wheel, brake pedal, and accelerator pedal were transmitted into the module. Eventually, the host vehicle in the scenario was controlled by the participants. In particular, the warnings were delivered to the driver in the form of a heads-up display (HUD) in the FCW validation test.

*3.2. Experiment Design*

For the driving characteristics acquisition test, twenty experienced drivers were recruited as experimental participants, including ten males and ten females. The ages of all the participants were between 24 and 36 years old (mean = 31, SD = 3.2). Each participant completed the test six times, the first of which was a pretest (data were not recorded). To evaluate the proposed FCW models, the same participants were recruited because all the classifiers were trained on the experimental longitudinal car-following data from particular drivers. The data-driven nature of this method ensures that the classifiers provide a forward collision warning that reflects the normal car-following strategy of the individual driver on whose data the classifiers were trained. The staged TTC, Honda, and Mazda models were also evaluated in the FCW validation test for references. After each experiment with a particular FCW model, experimenters were required to fill out a subjective questionnaire. Moreover, a baseline test was introduced.

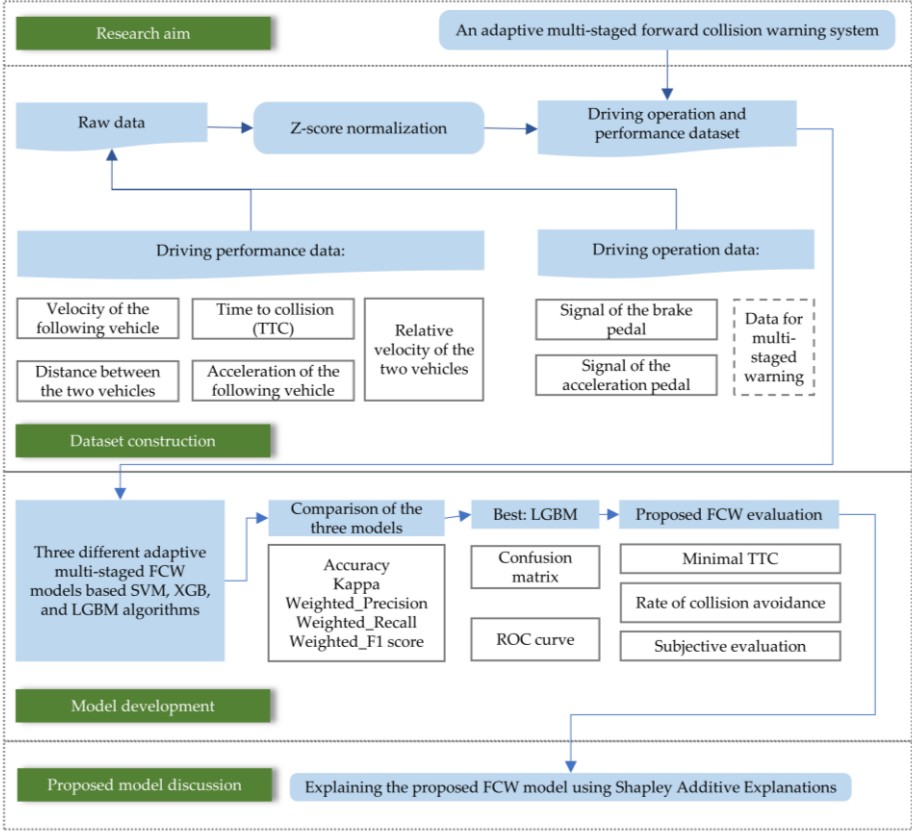

**Figure 1.** General framework.

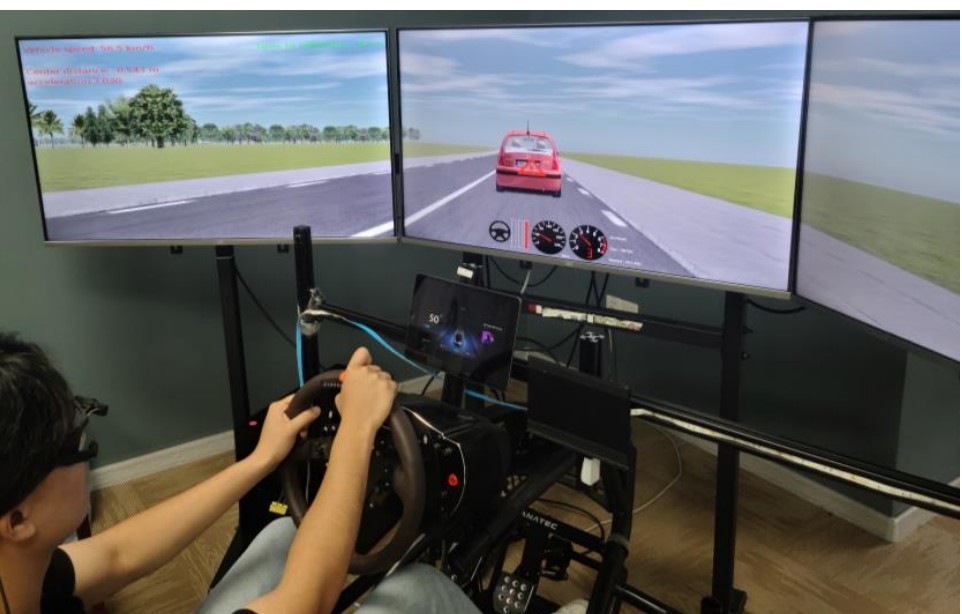

**Figure 2.** The open-cab driving simulator.

### 3.3. Data Collection and Variables

In the driving characteristics acquisition test, we focused on collecting and analyzing the data of driving performance. Five predictor variables were acquired at a sampling frequency of 20 Hz, namely, the velocity of the following vehicle, the relative velocities of the two vehicles, acceleration of the following vehicle, the distance between these two vehicles, and time to collision (TTC). Except for TTC, the four variables were widely used in FCW models based on machine learning and deep learning [9,24,27,31]. TTC was commonly used in perceptual approaches for FCW, which was designed to assume the time that would be taken for the crash risk between a preceding and a following vehicle [7]. As a result, TTC was utilized as a predictor variable because it characterized the driver's perception of the car-following incident. For the target variable (warning stage), it was defined by the frequently used driving intentions related to FCW (acceleration, braking) [8]. In the simulated driving experiments, we concluded that almost all the following vehicles in triggered FCW scenarios required urgent deceleration. The same results were also found in naturalistic driving experiments conducted on Chinese roads [31]. This means the driver needs to immediately brake to avoid a rear-end crash. Hence, the target variable (warning stage) was defined as follows: no warning stage (pressing the accelerator pedal), a first stage of warning (releasing the accelerator pedal: the driver believes there may be a risk of a collision ahead and releases the gas pedal), or a second stage of warning (pressing the braking pedal: then the driver presses the braking pedal to avoid the rear-end crash).

Figure 3 demonstrates the data of one triggered FCW sample. The start of the timing is when FCW is triggered by the front vehicle, and the end is the moment when TTC first increases to 3 s. TTC = 3 s is often used as the minimum FCW threshold [32,33]. Accordingly, we assume that when TTC increases again to 3 s, it indicates the end of the FCW event. For the target variable, zero is no warning stage, one indicates the first stage of warning, and two is the second stage of warning. All the predictor variables were standardized using z-score normalization.

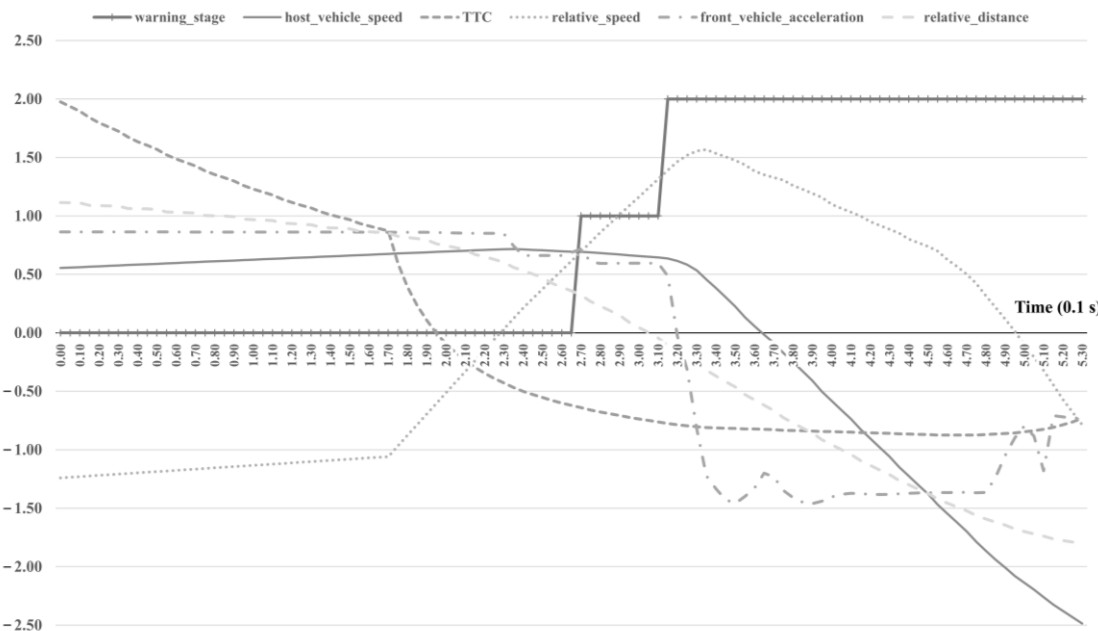

**Figure 3.** Example of recorded predictor variables and the target variable.

In the FCW validation test, we focused on evaluating the proposed model and then comparing it with other FCW models. For objective evaluation, minimal TTC and rate of collision avoidance were used, as these two metrics offer an effective way to evaluate the utility of FCW models [10,12,34,35]. Minimal TTC represents the average of the minimal TTC in all samples of the FCW validation test. The rate of collision avoidance represents the percentage of successful collision avoidance in all samples of a particular FCW model. In addition to objective indicators, subjective evaluation was also adopted to assess the proposed model. A subjective timing assessment is recognized as a useful measure to analyze the adaptability of FCW models [11,36]. The rating scales of FCW trigger timing were modified from Winkler's paper [11]. In addition, the assessment of the driver's acceptance of the model was performed based on the acceptance rating system of Van Der Laan (Figure 4) [37]. The above metrics were compared among staged TTC [33], Honda [18], Mazda [5], and the proposed model.

*3.4. Methodology*

Ensemble learning and machine learning algorithms were used to build adaptive FCW models because these algorithms could be useful for identifying and learning the driving behavior of a particular driver. Three algorithms were adopted, namely, a support vector machine (SVM), extreme gradient boosting (XGB), and a light gradient boosting machine (LGBM). These three algorithms were adopted and trained as multi-class classifiers. We introduced Optuna [38] to automate the hyperparameter searching of XGB and LGBM. GridSearchCV [6] was used for the hyperparameter optimization of the SVM. First, the five predictor variables from the previous section were input into the classifier. The classifier then output the results of the classification. As the algorithm of the FCW model, the classifier output different classification results corresponding to different warning stages of the FCW model. Finally, the model gave the driver the appropriate alerts based on the current warning stages. Accuracy, kappa, weighted F1-score, weighted precision, and weighted recall were then used to compare the performances of the above models. As a result, the most suitable algorithm was selected to build the FCW model. Finally, the output of the selected model was explained by the Shapley Additive Explanations (SHAPs) to measure the impacts of the five features on the results. Modeling and data analysis were performed using JupyterLab with the kernel of python 3.6.12. The three algorithms were used from scikit-learn 0.23.2.

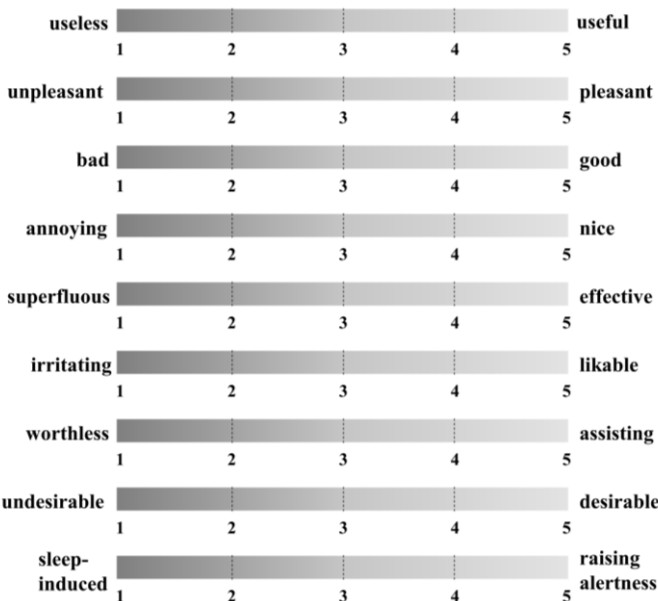

**Figure 4.** The subjective rating metrics of FCW trigger timing.

### 3.4.1. SVM

SVM is a supervised machine learning algorithm. The purpose of the SVM is to draw a line that "preferably" distinguishes between these two types of points so that if new points become available later, the line will also perform good classification. For multi-class classification, a SVC with an RBF kernel was used. The regularization parameter C was 0.9932524232943448, the degree of the polynomial kernel function was 3, the class_weight was "balanced", the cache_size was 5000, and the probability was "True". The other parameters were default values.

### 3.4.2. XGB

XGB is an algorithm implemented in the gradient boosting framework. It uses a pre-ranking method. The calculation process follows the ranking of feature values. The splitting gain of the current feature value is calculated by the sample of data. This way, the best splitting point can be found precisely. For the parameters of the XGB, n_estimators was 1000, learning_rate was 0.15097539992922396, max_depth was 10, subsample was 0.5, reg_alpha was 0, reg_lambda was 30, objective was "multi:softmax," and num_class was 3. The other parameters were default values.

### 3.4.3. LGBM

LGBM is a gradient boosting framework that uses a tree-based learning algorithm. It uses a histogram algorithm that discretizes continuous features into k discrete features and constructs a histogram of width k for statistical information (containing k bins). With the histogram algorithm, we do not need to traverse the data, but only k bins, to find the best splitting point. For the parameters of the LGBM, the objective was "multiclass," n_estimators was 1000, learning_rate was 0.27300747213500515, num_leaves was 140, max_depth was 4, lambda_l1 was 0, and lambda_l2 was 15. The other parameters were default values.

## 4. Results

### 4.1. Performance Comparison of the Three FCW Models

Figure 5 listed the comparison of the recognition accuracies of the different FCW models. LGBM had the highest prediction accuracy for the three warning stages. For the overall classification performance, LGBM and XGB were similar in accuracy, and the difference was not significant, but both were more accurate than the SVM model. As the

first stage had the smallest number of samples, the prediction accuracy for it was lower than in the other two stages.

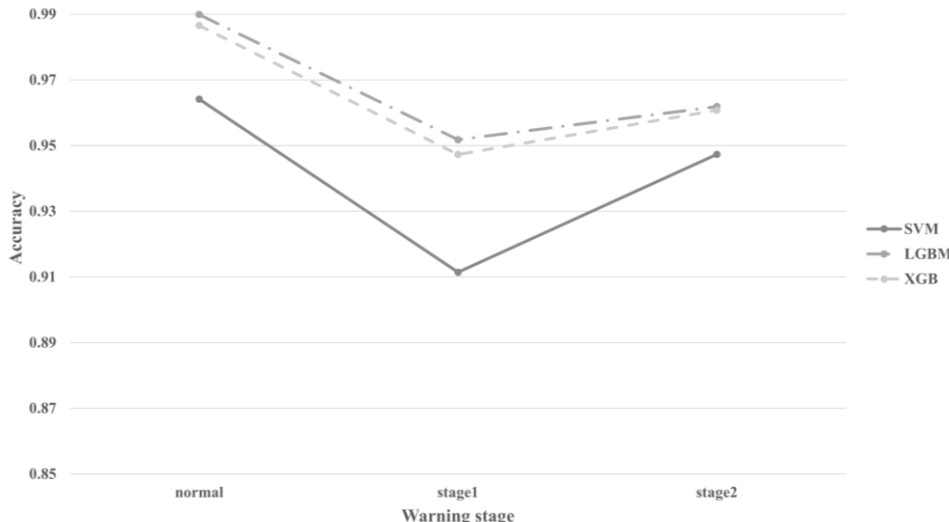

**Figure 5.** Accuracy at each stage of the FCW model based on three different algorithms.

Table 1 below shows five measures for the evaluation of the algorithms. The closer the value of these indicators is to one, the better the classification of the model is. The accuracy listed in Table 1 is the average accuracy of the three warning stages. Interestingly, the LGBM was observed to be the most accurate model. Due to the aforementioned imbalance in the sample category, kappa was used to evaluate the models so as to avoid the bias in accuracy caused by a large proportion of samples being in one category. An unexpected result was then found, in that LGBM had the highest kappa (0.91552) among the three models. The weighted F1-score was also used as a balanced indicator of precision and recall. Generally, LGBM performed the best in all measures in Table 1; therefore, it was selected as the algorithm for the proposed FCW model.

**Table 1.** Comparison of the performances of the three algorithms.

|  | **LGBM** | **XGB** | **SVM** |
|---|---|---|---|
| Accuracy | 0.96786 | 0.96487 | 0.94096 |
| Kappa | 0.91552 | 0.90762 | 0.84838 |
| Weighted_Precision | 0.95192 | 0.94691 | 0.92440 |
| Weighted_Recall | 0.95179 | 0.94731 | 0.91143 |
| Weighted_F1 score | 0.95185 | 0.94707 | 0.91632 |

*4.2. LGBM-Based FCW Model Assessment*

Figure 6a shows the validation results of the LGBM model using the confusion matrix based on the result of the LGBM's prediction of the acquired data. Moreover, the receiver operating characteristic (ROC) curve is demonstrated in Figure 6b. As can be seen in the figure, the results indicate that the proposed model was successfully validated, since the area under curve (AUC) of the micro-average ROC curve was estimated as 0.989.

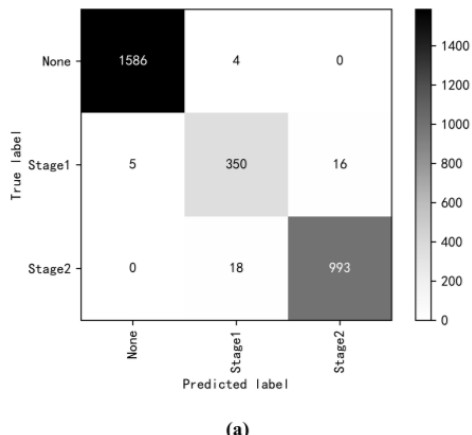
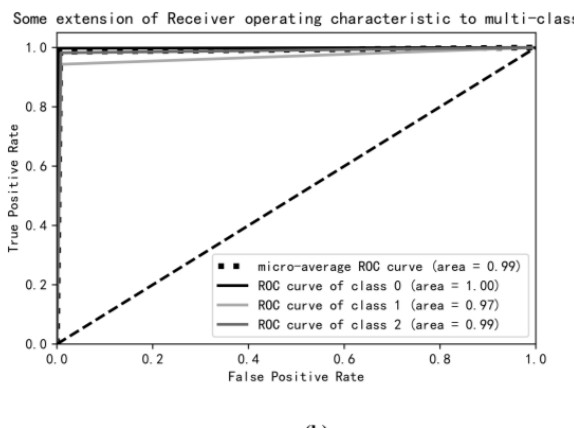

<div align="center">(a)         (b)</div>

**Figure 6.** The assessment results of the proposed model expressed in terms of the confusion matrix (**a**) and the ROC curve (**b**).

### 4.3. Proposed FCW Validation

After a defined emergency braking event of the front vehicle, the driver in the following vehicle brakes to avoid a rear-end collision. During the above process, a minimal TTC will exist. Figure 7a shows the comparison of minimal TTC. The staged TTC model had the biggest minimal TTC, followed by the proposed model. Though the Honda model had the lowest mean minimal TTC among the four models, its mean minimal TTC was still higher than that of the baseline. As shown in Figure 7b, the proposed and staged TTC model helped the participants avoid all the collisions, whereas the Honda model had the smallest collision avoidance rate.

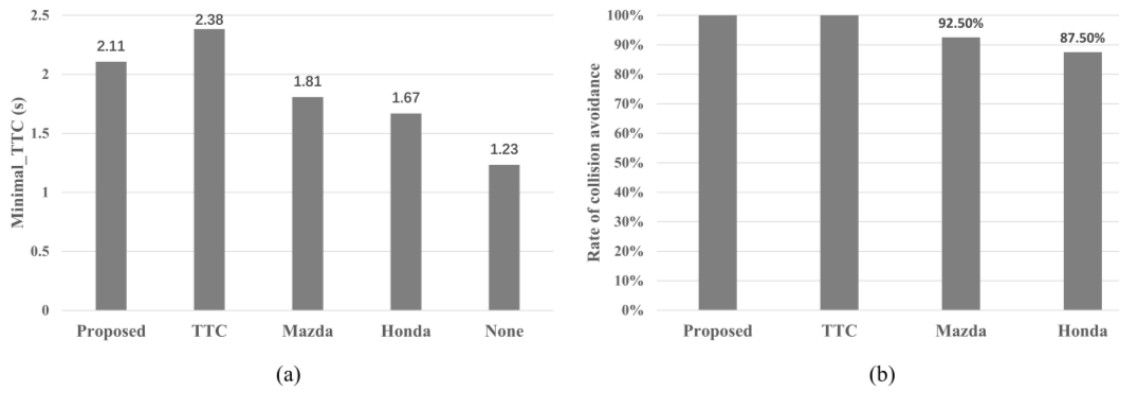

<div align="center">(a)         (b)</div>

**Figure 7.** The objective utility of the proposed model expressed in terms of the minimal TTC (**a**) and the rate of collision avoidance (**b**).

As shown in Figure 8, we used the participants' subjective evaluations of the four FCW models in the validation test. The proposed model had the most appropriate timing among the four models. The timing of the staged TTC model and that of the Mazda model were considered by the participants to be slightly earlier, whereas the timing of the warning from the Honda algorithm was considered late. For user acceptance, the proposed model had the highest acceptance rate, whereas the rate of the staged TTC model was the lowest.

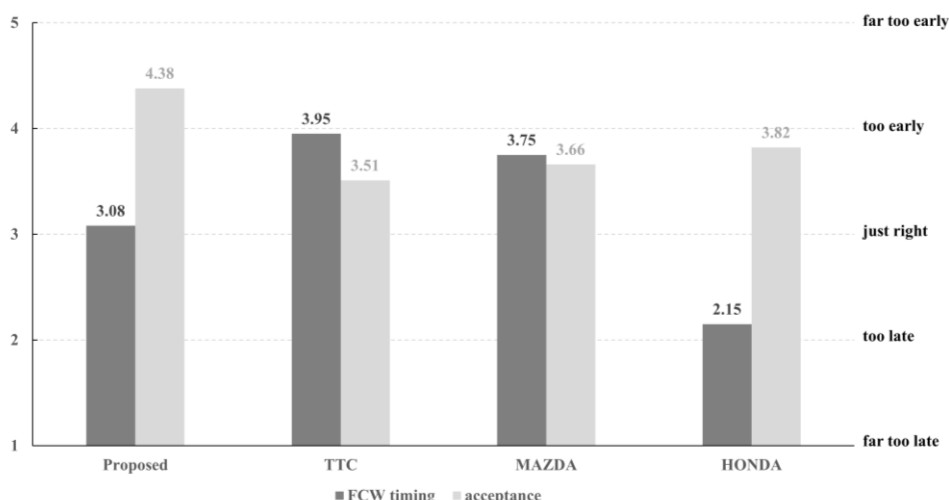

**Figure 8.** The subjective utility expressed in terms of the FCW timing and acceptance of the proposed model.

## 5. Discussion

We proposed an adaptive multi-staged FCW based on the LGBM in the present study. For the adaptivity of the proposed model, the LGBM model was trained on the dataset of the user's driving styles. Therefore, the proposed FCW model is capable of learning the preferences of car-following habits and the perception of dangers from the specific user. Furthermore, two staged warnings according to the emergency of the forward collision are given to the driver to minimize false warnings and nuisance warnings. The results indicated (Figure 5, Table 1) that the proposed LGBM was able to accurately recognize all the warning stages. As shown in (Figure 5, Table 1), the recognition accuracy in the three warning statuses of SVM was the lowest. Although SVM has a strong ability to handle small sample classification, it is not suitable for multi-classification. This undesirable result can be partly explained by the fact that the target is multi-class and unbalanced samples. The overall recognition accuracies of the XGB and LGBM were similar. However, the LGBM had higher accuracy in the first warning stage (95.179%). This result could be attributed to the stratified k-fold method used in the LGBM for cross-validation. It is based on stratified random sampling, and it ensures that the label category distribution of each cross-validation data remains the same as the original sample when each category sample is unbalanced [39]. In general, therefore, it seems that the FCW model built with the LGBM is able to perform well even with unbalanced samples in the target variable.

As previously mentioned, the LGBM is an ensemble learning algorithm, but is difficult to explain, being like a black box. However, the relationship between the model's output and features (five predictor variables) can be analyzed and then explained by Shapley Additive Explanations (SHAPs). The SHAP value is the average marginal contribution of a feature value across all possible coalitions [40]. Figure 9 demonstrates the average contributions of the predictor variables to the output of the model. Among all features, the front vehicle acceleration had the largest contribution, whereas the distance between the two vehicles had the smallest contribution to the classification. For the no-warning stage, the front vehicle acceleration and TTC are the two significant features. Concerning the second warning stage, the front vehicle acceleration and relative velocity are the two features that mainly affect the characteristics. Surprisingly, the presence of the two features, the delta velocity and TTC, can help distinguish between these two stages. As mentioned before, although the distance between the two vehicles was the feature with the least contribution, this feature was able to distinguish the first warning stage from the other two stages. In summary, the contributions of the features to the classification results make the FCW algorithm interpretable and justify the selected feature metrics as well.

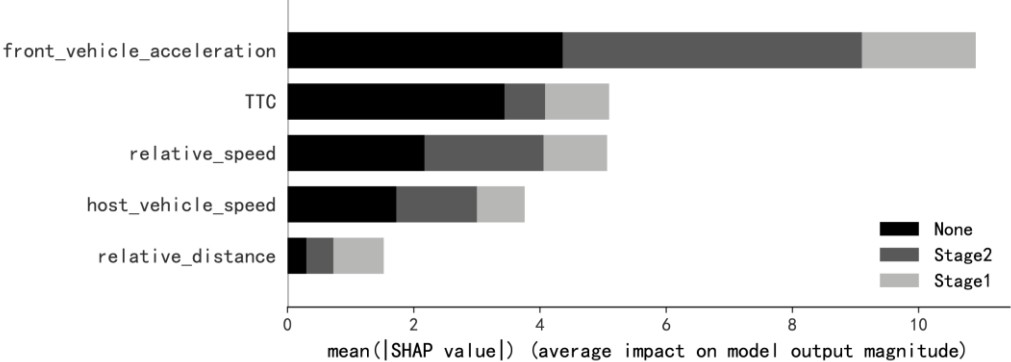

**Figure 9.** Results of the SHAP summary plot.

In the validation test, two objective indicators were used to evaluate the utility of FCW models. As demonstrated in Figure 7, in general, the proposed model that provides staged warnings is an effective FCW proposal. Firstly, the collision avoidance rate was adopted as a straightforward indicator for assessment. According to Figure 7b, the Mazda model had a higher rate of collision avoidance than the Honda one. This result is consistent with data obtained in the earlier study [10]. In contrast to earlier findings, however, the TTC model had a higher rate of collision avoidance than the Mazda model. This discrepancy could be attributed to the staged warnings of the TTC model in our study. It can thus be suggested that multiple stages of warning could improve the utility of the model. Secondly, minimal TTC indicates the situation when the risk of collision is at its highest. Therefore, the risk of collision decreases with its value. Surprisingly, the minimal TTC of the proposed model was found to be a bit lower than that of the staged TTC model. However, it cannot be concluded from this result that staged TTC is better than the proposed model, as the driver adaptability of the model has not been taken into account.

A subjective utility assessment was adopted to evaluate the adaptability of the FCW models to the driver's style of driving. As shown in Figure 8, the proposed model had the most suitable warning timing, staged TTC had the earliest warning timing, and the Honda model had the latest timing. As previously discussed, although the minimal TTC of the staged TTC model is the largest among the four models, its timing is fixed and was too early for the participants. Previous research also suggests that a too-early warning makes the drivers perceive it to be a false or nuisance warning [13]. In addition, combined with the result of acceptability, a trade-off was found between utility and user acceptance. Specifically, on one hand, in order to improve the utility, the warning timing of the FCW model should be early enough so that the driver has enough time to take appropriate measures to avoid the danger. On the other hand, too-early warnings can interfere with the driver's normal driving, reduce driving comfort, and finally, lead to lower user acceptance. This could be used to explain the lowest user acceptance of the staged TTC model, despite its high utility. As a result, the balance between the FCW trigger timing and user acceptance should be emphasized. Since each driver has a different driving style, proficiency, and perception of following distance [41], FCW is designed not only to provide staged warnings to drivers, but also to learn driving characteristics so as to provide the most appropriate timing of FCWs.

In this study, we only developed and evaluated the proposed FCW model using driving simulators. We also consider it an important task to compare the model's difference in performance in simulated driving and real-life driving. In a future study, we plan to develop our model based on naturalistic driving tests and controlled field tests. The repeated-training interval will be estimated and validated as well. Moreover, neural network algorithms will be considered in the comparison study. We also found an interesting relationship between the accuracy of the model and the warning stage: the fewer the warning stages, the higher the classification accuracy. This finding may be attributed to the fact that as the number of warning stages increases, the differences between corresponding

predictor variables in different stages are not obvious enough. This leads to the lower accuracy of the classifier. If the number of warning stages is smaller, the differences between corresponding predictor variables are obvious, and the classification accuracy of the model is higher. To further validate this conjecture, more designs of warning stages and models are needed. However, for more warning stages (more than the two stages in this article), it is indeed a difficult task to label the target variables. Therefore, we will consider the effect of different warning stages on FCW model performance in future research.

## 6. Conclusions

In the present study, we proposed a staged adaptive FCW system which utilizes V2X communication to transfer data between the front and the following vehicle. A light gradient boosting machine (LGBM) was proposed to provide a staged warning of possible collision with the front car. The results of the driving characteristics acquisition test indicated that the proposed LGBM exhibited high accuracy for recognizing the warning stages. The results of the validation test demonstrated that compared with the traditional FCW models, the proposed FCW model provides a more effective and adaptive staged warning to the driver. As a result, the proposed FCW model made the FCW system safer, and more acceptable for the driver due to the absence of annoying warnings.

**Author Contributions:** J.M.: Conceptualization, Supervision. J.L.: Formal Analysis, Investigation, Writing—Original Draft, Writing—Review and Editing, Methodology. Z.G.: Resources, Investigation. H.H.: Investigation, Data Curation. All authors have read and agreed to the published version of the manuscript.

**Funding:** This work was Supported by the Shanghai Municipal Science and Technology Major Project (2021SHZDZX0100) and the Fundamental Research Funds for the Central Universities.

**Institutional Review Board Statement:** The study was conducted in accordance with the Declaration of Helsinki, and approved by the Science and Technology Ethics Committee of Tongji University (tjdxsr012) for studies involving humans.

**Informed Consent Statement:** Informed consent was obtained from all subjects involved in the study.

**Data Availability Statement:** Not applicable.

**Conflicts of Interest:** On behalf of all the authors, the corresponding author states that there is no conflict of interest.

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
