# Peer review of "An Adaptive Multi-Staged Forward Collision Warning System Using a Light Gradient Boosting Machine"

_information, doi:10.3390/info13100483_

Round 1
Reviewer 1 Report
The paper is technically sound, however:
1. There is no literature review section whereby the authors should discuss relevant work and the position of their paper amongst seminal works.
2. why did you choose these ML algorithms? and how did you select the hyperparameters for them?
3. Also, I would like to see a schematic diagram that shows the whole framework and steps followed in the paper.
Reviewer 2 Report
The reviewed work is very interesting. The authors present innovative approaches to the problem. Two aspects should be improved in the work: - firstly, the reference to the state of the art of world literature is insufficient. It is essential to extend the literature review. - secondly, what is more important, is basing your analyzes only on the results carried out in the driving simulator. From my experience and from the many studies I have participated in, the correlation between simulator research and real life research is very little or not at all. In order for the work to be complete, the authors should compare the tests performed on the simulator with the tests performed in the real environment, e.g. on the test track. Meaningful conclusions can be drawn from such a comparison. If only simulator tests are performed, the work is partially done and does not correspond to reality.
Round 2
Reviewer 1 Report
Authors have addressed my concerns
Author Response
Dear Reviewer,
Thank you very much for your time involved in reviewing the revised manuscript and your very valuable suggestions.
Sincerely,
The Authors
Reviewer 2 Report
Currently the work is sufficient for publication. However, I emphasize once again in subsequent studies the simulator itself is not enough. Driver behavior in the simulator differs from real driving behavior. it is true that there is some correlation, but the results differ significantly. It all depends on the accident situation (scenario). I recommend that you analyze the items of literature:
The test methods and the reaction time of drivers, Eksploatacja i Niezawodność - Maintenance and Reliability, ISSN: 1507-2711
Driver model for the analysis of pre-accident situations, Vehicle System Dynamics, ISSN: 0042-3114
Author Response
Dear Reviewer,
Thank you very much for your time involved in reviewing the revised manuscript and your very valuable comments. We are very sorry for the lack of on-road tests and we plan to study the drivers' behavior in the real environment. The literature you recommended to us is of great help to build a driver model in a real scene. According to the important finding from the literature, the dependence of driver reaction times in connection with the turning, engine braking, and service brake braking maneuvers and risk time will be taken into consideration in our future studies. Thanks for the comments and literature that enlightened us.
Sincerely,
The Authors